# Scientific progress made towards bridging the knowledge gap in the biology of Mediterranean marine fishes

Eva Daskalaki[1]*, Evangelos Koufalis[1], Donna Dimarchopoulou[2,3], Athanassios C. Tsikliras[1]

1 Laboratory of Ichthyology, Department of Zoology, School of Biology, Aristotle University of Thessaloniki, Thessaloniki, Greece, 2 Department of Biology, Dalhousie University, Halifax, Nova Scotia, Canada, 3 Department of Biology, Woods Hole Oceanographic Institution, Woods Hole, Massachusetts, United States of America

* daskalakieva2@gmail.com, evangeld@bio.auth.gr

**Data Availability Statement:** All relevant data are available from review articles cited within the paper and at www.fishbase.org.

## Abstract

The Mediterranean Sea is a renowned biodiversity hotspot influenced by multiple interacting ecological and human forces. A gap analysis on the biology of Mediterranean marine fishes was conducted in 2017, revealing the most studied species and biological characteristics, as well as identifying knowledge gaps and areas of potential future research. Here, we updated this gap analysis five years later by reviewing the literature containing information on the same eight biological characteristics, namely length-weight relationships, growth, maximum age, mortality, spawning, maturity, fecundity and diet, for the 722 fish species of the Mediterranean Sea. The results revealed a considerable knowledge gap as 37% of the species had no information for any of the studied characteristics, while 13% had information on only one characteristic. Out of all the biological characteristics, the smallest knowledge gap was found in the length-weight relationships (studied for 51% of the species, mainly in the eastern Mediterranean), while the least studied characteristic was mortality (studied for 10% of the species). The western and eastern Mediterranean Sea were leading forces in data collection exhibiting the narrowest gaps between current and desired knowledge. The most studied species across the entire region were the highly commercial European hake (*Merluccius merluccius*), red mullet (*Mullus barbatus*), European anchovy (*Engraulis encrasicolus*), European pilchard (*Sardina pilchardus*), common pandora (*Pagellus erythrinus*), and annular seabream (*Diplodus annularis*). The knowledge gap has shrunk by 6% during the last five years, with 40 new species having at least one study on their biology. Moreover, research has slightly shifted towards species that have been traditionally neglected, e.g., sharks, rays and chimaeras (chondrichthyans). It is recommended that research becomes less focused on commercial species and more targeted towards the identified gaps, vulnerable species (e.g., deep-sea species and chondrichthyans) and species that could potentially pose a threat (e.g., non-indigenous species) to the ecosystems of the everchanging Mediterranean Sea.

**Funding:** This research was partly funded by the European Union's Horizon 2020 Research and Innovation Program (H2020-BG-10-2020-2), grant number No. 101000302 - EcoScope (Ecocentric management for sustainable fisheries and healthy marine ecosystems). The funders had no role in study design, data collection and analysis, decision to publish, or preparation of the manuscript.

**Competing interests:** The authors have declared that no competing interests exist.

## Introduction

The Mediterranean Sea has been identified as an area with a large information gap on the biological characteristics of fish species [1]. The lack of biological information impedes the efforts of scientists working in different fields, as this information is needed for stock assessments and ecosystem models. Fish stock assessment models alone require an array of biological information such as length-weight relationships, maximum age, size at maturity, and growth parameters [2], leading scientists to develop new and less data-demanding methods for estimating the desired parameters [3]. In addition to assessment models, life history traits such as maximum age, growth rate and age at maturation, affect species resilience against overexploitation [4, 5] and climate change [6], therefore playing an important role in modelling species resilience. Moreover, the more holistic approach of the ecosystem-based fisheries management (EBFM) requires biological data on all ecosystem components, both commercial and non-commercial, along with the characteristics of individual species, to be taken into consideration for the decision-making process [7, 8].

As a recognized global biodiversity hotspot, the Mediterranean Sea needs to be protected [9, 10]. Despite the fact that there are established marine protected areas (MPAs) around the Mediterranean basin covering 8.33% of its surface, the cumulative surface of no-go, no-take or no-fishing area represents only 0.04% of it [11]. Therefore, only 0.23% of the Mediterranean basin is actually and effectively managed and protected [12], alongside the species living in these areas. Given the importance of data in systematic management, the reported lack of data regarding Mediterranean biodiversity [1, 9, 13] is a handicap to any conservation effort.

In the gap analysis conducted by Dimarchopoulou *et al.* (2017) [1] on the biological characteristics of Mediterranean marine fishes, sharks, rays and chimaeras (chondrichthyans) were identified as one of the least studied groups in the Mediterranean region, with scarce information available for only a handful of species. Many of these species are top predators in the environments they inhabit and as such, they play a key role in the food web through direct and indirect cascading effects [14, 15]. Additionally, chondrichthyans are characterized by low fecundity, late maturity and slow growth and are exposed to considerable fishing pressure all over the world, which makes them extremely vulnerable [16]. Although chondrichthyans are not explicitly targeted in the Mediterranean region, they are often caught as bycatch [17] and research has shown that Mediterranean elasmobranchs (sharks and rays) are either of poor conservation status compared to the rest of the world or simply neglected [17, 18]. Furthermore, historically declining trends in the abundance of chondrichthyan species have been reported in the Mediterranean region. For some species like hammerhead *(Sphyrna spp.)*, blue shark *(Prionace glauca)*, Shortfin mako *(Isurus oxyrinchus)*, Porbeagle *(Lamna nasus)*, *and* thresher sharks *(Alopias vulpinus)* the decline in abundance has been estimated between 96 and 99.99% in the western Mediterranean region [19]. At the same time, the fishing effort is evidently becoming more intense, pushing these vulnerable species to their limits [15, 20, 21].

Other Mediterranean fish species adding to the Mediterranean's unique biodiversity are deep-sea species. This diverse group of species (Actinopterygii and Elasmobranchii) inhabits deep-sea ecosystems covering 79% of the entire basin [22] and is found beneath approximately 200 m depth [23]. Deep-sea species are particularly vulnerable due to "K-selected" life history characteristics, as a consequence of the low productivity environment they inhabit [24]. Research has revealed that Mediterranean deep-sea biodiversity is richer than once expected, and yet still largely undiscovered and understudied [23] due to intricate and laborious logistics required for its exploration [25]. Nevertheless, fishing [26] and various maritime exploitative activities have been reportedly pushed towards larger depths on a global scale [27] and especially in the Mediterranean region [22], as the fish stocks of the continental self are being

depleted [28, 29]. The lack of data regarding basic biological characteristics of many deep-sea species makes implementation of effective management regulation for these vulnerable and important ecosystems a challenging task [30].

Moreover, the Mediterranean region is facing rapid changes including acidification [31, 32] climate-driven warming [33] and the influx of non-indigenous species [34]. Non-indigenous species that establish reproductive populations become invasive species [35], posing a threat to human health [36], local populations [37] and the economy [34, 38], additionally to the effects on the ecosystem, as a whole [39]. The rate at which the invasion is taking place is alarming [40], thus gathering data on the biological characteristics of non-indigenous species in the Mediterranean region, is of major importance in order to identify, monitor and predict the behavior and distribution of these species and successfully integrate non-indigenous species into an ecosystem-based management approach over the entire basin [41].

Considering that five years have passed since the initial gap analysis by Dimarchopoulou *et al.* (2017) [1] and given 1) the numerous stressors co-occuring in the Mediterranean region, 2) the imperative need for monitoring and efficient management, activities that require all-encompassing ecosystem modelling and data availability and 3) the reported knowledge gap in fundamental biological characteristics of marine fish, we revisited the topic to keep track of the existing gap and identify any progress made. Finally, we aim to provide some updated recommendations pointing at the existing gaps as priority targets for future scientific projects and put a spotlight on groups of species of great value to the Mediterranean biodiversity and therefore, of conservation interest (deep-sea species and chondrichthyans) as well as species described in the literature as potential stressors for the ecosystems (invasive non-indigenous species).

## Materials and methods

Following the methodology of Dimarchopoulou *et al.* (2017) [1] we updated the gap analysis on the biology of Mediterranean marine fish species, taking into consideration the Mediterranean Sea as a whole, but also analyzing the data on a subregional basis (W: western; C: central; E: eastern, Mediterranean). The gap between the current and desired knowledge was investigated for the same eight biological characteristics, i.e., length-weight relationships, growth, maximum age, mortality, spawning, maturity, fecundity and diet. The desired knowledge level has been defined as *having information on most biological characteristics for at least half of the Mediterranean marine fishes* [1].

The updated gap analysis performed here covers the years 2015–2021 and investigates the relevant studies on the biology of Mediterranean marine fishes that have been published after the original gap analysis. We took into consideration the two-year literature overlap (2015–2016) with the initial gap analysis by excluding the papers published in 2015 and 2016 that had already been included in the Dimarchopoulou *et al.* (2017) paper [1]. For our analysis, we collected information on all fish species that have been recorded in the Mediterranean Sea large marine ecosystem as they are listed in FishBase [42]. Specifically, we recorded the existence (or not) of papers containing information on eight biological characteristics for every species in the list (S1 Appendix). Altogether, 758 Mediterranean marine fish species were listed in Fish-Base [42], of which 36 species were excluded as misidentified and/or questionable records. The current list of Mediterranean fish species is not identical to the one used in Dimarchopoulou *et al.* (2017) [1]. Our updated list consists of 722 species, while the first gap analysis used a list of 714 species. Apparently, there have been a few changes in the FishBase Mediterranean species list [42], during the past five years. Nevertheless, the changes regarding the fish species are minor: 709 species are the same in both lists, 5 species existed only in the Dimarchopoulou

*et al.* (2017) [1] list but have been removed from the Mediterranean fish species list in FishBase most probably to identified synonyms [42], while 13 new species were added in that list since the first gap analysis.

For each of the 722 analyzed species, the available information on length-weight relationships, growth parameters, maximum age, mortality rate, spawning period, size at maturity, fecundity and diet composition was extracted from FishBase [42] and published literature that was searched through SCOPUS. For the length-weight relationships (LWR) we recorded species with information on both the slope (b) and intercept (a) of the equation; for somatic growth (G) we considered records with the asymptotic length ($L_\infty$) and the rate at which $L_\infty$ is approached (K), leaving out all growth records of "questionable" status in FishBase [42] and for the lifespan we included records with the maximum age ($t_{max}$). Concerning the reproduction parameters, we considered the onset and duration of spawning (Spawn) and length at maturity ($L_m$) to identify spawning and maturity related information, respectively. Additionally, we regarded absolute (the actual number of eggs) and relative (the number of eggs per unit of weight) number of oocytes for fecundity (Fec) [43]. Lastly, we considered as records for diet papers that included information on prey items, stomach content and feeding preferences as records for diet, while the natural mortality rate was used as natural mortality (M), regardless of the estimation method.

Altogether, 444 papers published between 2015 and 2021 were screened, i.e., 35 and 409 papers extracted from FishBase [42] and SCOPUS, respectively (S2 Appendix). We did a search for each Mediterranean fish species in FishBase [42] and extracted information coming from the Mediterranean Sea on the eight studied characteristics which are presented in the "More information" section of each species page. Regarding the SCOPUS literature review, we conducted a search using the scientific name of each fish species in our list alongside the name of the region, e.g., "*Merluccius merluccius*" AND "Mediterranean". We then went over the title and abstract of each search result to identify the ones that met the aforementioned criteria and could be included in this review.

To accommodate our research, we divided fish species into categories: commercial, non-commercial, deep-sea (species that live in depths of 200m and below), non-indigenous (species that have been introduced into the Mediterranean ecoregion), chondrichthyans (sharks, rays and chimaeras), protected, and species with atypical life history strategies (e.g., those exhibiting very slow growth or providing parental care to their offspring). Regarding protection, we recorded the protection status (IUCN Red List of Threatened Species) of each species based on the IUCN categories (LC: least concern; EN: endangered; DD: data deficient; NE: not evaluated; NT: near threatened; VU: vulnerable; CR: critically endangered), alongside their commercial value (Val), which was shown as "price category" in FishBase [42] (VH: very high; H: high; M: medium; L: low) and their trophic level, as it was calculated in FishBase [42]. Furthermore, we used the depth range of each species provided by FishBase [42], to discern which fish species inhabit the deep-sea ecosystems of the Mediterranean Sea. The deep-sea zone is considered to begin from 200m underneath the surface [23], so we included all the fish species with depth ranges starting from around 200 meters and deeper. We also extracted the Mediterranean Sea catches averaged for the years 2015–2019 from the FAO-GFCM database [44]. We chose to keep only the species-specific catch data and not the catches assigned to higher taxonomic groups (e.g., genera, families).

At last, we cross-correlated the number of records per species with the number of total catches, the maximum reported somatic length ($L_{max}$) and the trophic level for species with at least one record and available data (total catches: n = 125, $L_{max}$: n = 451 and trophic level: n = 452), to detect any bias of the scientific effort towards commercial species, larger species or species of higher trophic level, respectively. We started by checking the normality of our data,

using the Shapiro–Wilk test of normality and then we proceeded to search for linear correlation between each one of the three variables and total catches using the Pearson's correlation coefficient (r) for the parametric data and Spearman's correlation coefficient (ρ) for the non-parametric data. Our data were logarithmically transformed to improve the visualization process. Finally, following the Dimarchopoulou *et al.* (2017) [1] methodology, we calculated the number of records per metric ton of catch (records/t) to allow comparisons between highly commercial and non-commercial species and per centimeter of somatic length (records/cm) to allow comparisons among sizes.

## Results

Based on the present analysis, there is no information on any biological characteristic for 270 out of the 722 Mediterranean fish species (37%), while for 95 (13%) of them there is information for only one characteristic. Regarding the biological characteristics separately, the gap is narrower for length-weight relationships, as they have been studied for 366 (51%) species, followed by spawning (312 species; 43%), diet (284 species; 39%), growth (211 species; 29%), maturity (192 species; 27%), maximum age (184 species; 25%) and fecundity (142 species; 20%) (Fig 1, *top row*). Most new studies focused on length-weight relationships, diet and spawning, with 708 new length-weight relationship records for 56 species, 467 new diet records for 76 species and 190 new spawning records for 34 species (Table 1) in the last five years.

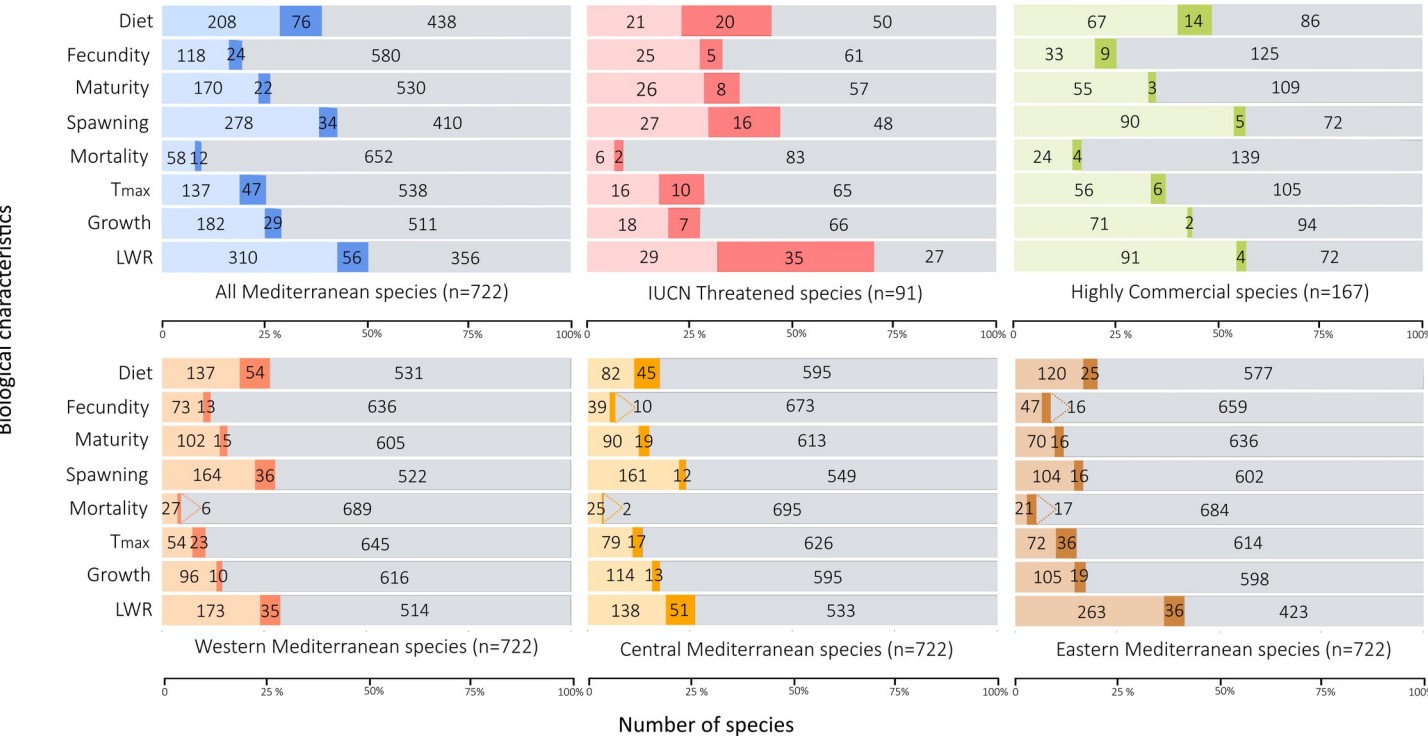

**Fig 1. Numbers (and percentages) of Mediterranean fish species with published information on eight biological characteristics, namely, length-weight relationships (LWR), growth parameters (G), maximum age (t_max), mortality rate (M), spawning period (Spawn), size at maturity (L_m), fecundity (Fec) and feeding preferences (Diet).** *Top row*: blue colour: all Mediterranean fish species (top left panel), red colour: Mediterranean fish species listed in the IUCN Red List of Threatened Species under the categories near threatened (NT), vulnerable (VU), endangered (EN) and critically endangered (CR) (top middle panel), green colour: highly commercial Mediterranean fish species under the categories high (H) and very high (VH) commercial value in FishBase (top right panel). *Bottom row*: orange colour: Western Mediterranean fish species (bottom left panel), yellow colour: central Mediterranean fish species (bottom middle panel) and brown colour: eastern Mediterranean fish species (bottom right panel). Light coloured data are from Dimarchopoulou *et al.*, (2017) [1], while darker coloured data are from the present study and data coloured in light gray represent the number of fish species lacking published information on the respective biological characteristic.

**Table 1. The number of records from Dimarchopoulou *et al.* (2017) [1] and from the papers published between 2015–2021, as well as the total number of records from the present study for each one of the studied characteristics.**

| Characteristic | Number of records from Dimarchopoulou *et al.* (2017) [1] | Number of records published from 2015 to 2021 | Total number of records_Present study |
|---|---|---|---|
| Diet | 563 | 467 | 1030 |
| Fecundity | 184 | 62 | 246 |
| Maturity | 393 | 143 | 536 |
| Spawning | 670 | 190 | 860 |
| Mortality | 85 | 37 | 122 |
| $T_{max}$ | 283 | 187 | 470 |
| Growth | 691 | 132 | 823 |
| LWR | 1738 | 708 | 2446 |

$T_{max}$: maximum age; LWR: length-weight relationships

The largest gap was found in natural mortality for which information was sparse (70 species; 10%) (Table 1). Concerning the studied species that are listed under the categories near threatened (NT), vulnerable (VU), endangered (EN) and critically endangered (CR) of the IUCN Red List (n = 91), the percentages for all biological characteristics except mortality and growth were higher compared to the total of the species (Fig 1, *top row*). In terms of commercial value, the species listed under the categories high (H) and very high (VH) (n = 167), had higher percentages in every biological characteristic, compared to all species listed (Fig 1, *top row*).

With respect to the studied characteristics, the record distribution pattern was consistent among the western, central and eastern Mediterranean, with the exception of some spatial variations within each biological characteristic (Fig 1, *bottom row*). Spawning, maturity, fecundity and diet were most extensively studied in the western subregion, while length-weight relationships and maximum age were studied mostly in the eastern subregion. Growth and mortality records were rather evenly distributed among the three subregions (Fig 1, *bottom row*).

In terms of number of records, the most studied species were European hake *Merluccius merluccius* (227 records in total with 46 new records in the last five years), red mullet *Mullus barbatus* (163 records in total with 34 new), European anchovy *Engraulis encrasicolus* (142 records in total with 55 new), European pilchard *Sardina pilchardus* (125 records in total with 43 new), common pandora *Pagellus erythrinus* (124 records in total with 26 new), annular seabream *Diplodus annularis* (115 records in total with 17 new), surmullet *Mullus surmuletus* (115 records in total with 15 new) and bogue *Boops boops* (100 records in total with 24 new) (Table 2). Furthermore, we identified many new records for chondrichthyans, resulting in some chondrichthyan species with more than 30 records, although there still are 13 species (out of 82 total chondrichthyan species) without a single record on their biological characteristics (Table 3).

The least studied species were those belonging to the Blennidae, Gobiesocidae, Gobiidae, Moridae, Myctophidae, and Syngnathidae families (Table 4), as well as to two groups of fish species: non-indigenous species and deep-sea species (Figs 2 and 3). There are 126 non-indigenous species in FishBase [42] under the "introduced" status, from the list of 722 Mediterranean fish species (17.5%). For 60% of these non-indigenous fish species there is no available record on the specific biological characteristics studied in this work. Additionally, for 15% of them, information exists on one out of eight biological characteristics (Fig 2). Only eight non-indigenous fish species can be considered well studied (Goldband goatfish *Upeneus moluccensis*, Por's goatfish *Upeneus pori*, Redcoat *Sargocentron rubrum*, Shrimp scad *Alepes djedaba*, Silver-cheeked toadfish *Lagocephalus sceleratus*, Devil firefish *Pterois miles*, Dusky spinefoot *Siganus*

**Table 2. List of the most studied fish species in the Mediterranean Sea based on the number of records (Number of Rec.), the number of studied characteristics (Number of Char.), and the number of records per characteristic.**

| Species | Common Name | Family | IUCN | Commercial Value | Number of Rec. | Number of Char. | Number of records per characteristic_Present study |
|---|---|---|---|---|---|---|---|
| *Merluccius merluccius* | European hake | Merlucciidae | LC | H | 227 | 8/8 | 59 LWR, 58 G, 29 Diet, 24Sp, 23 Mat, 15 A, 10 Fec, 9 M |
| *Mullus barbatus* | Red mullet | Mullidae | LC | M | 163 | 8/8 | 49 LWR, 47 G, 19 Diet, 13 A, 12 Sp, 12 Mat, 7 Fec, 4 M |
| *Engraulis encrasicolus* | European anchovy | Engraulidae | LC | M | 142 | 8/8 | 39 LWR, 23 Sp, 23 G, 23 Diet, 13 A, 10 Mat, 7 Fec, 4 M |
| *Sardina pilchardus* | European pilchard | Clupeidae | LC | L | 125 | 8/8 | 29 G, 24 LWR, 22 Diet, 19 Sp, 11 Mat, 10 A, 5 Fec, 5 M |
| *Pagellus erythrinus* | Common pandora | Sparidae | LC | M | 124 | 8/8 | 49 LWR, 17 G,15 A, 14 Diet, 13 Sp, 12 Mat, 3 Fec, 1 M |
| *Mullus surmuletus* | Surmullet | Mullidae | LC | VH | 115 | 7/8 | 48 LWR, 18 Sp, 17 G, 14 Diet, 8 Mat, 7 A, 3 M, 0 Fec |
| *Diplodus annularis* | Annular seabream | Sparidae | LC | L | 115 | 8/8 | 57 LWR, 17 Sp, 13 G, 8 A, 7 Mat, 7 Diet, 4 Fec, 2 M |
| *Boops boops* | Bogue | Sparidae | LC | H | 100 | 8/8 | 36 LWR, 19 G, 15 Sp, 10 A, 9 Mat, 5 Diet, 3 M, 3 Fec |
| *Diplodus vulgaris* | Common two-banded seabream | Sparidae | LC | L | 80 | 8/8 | 30 LWR, 14 Diet, 10 Sp, 8 G, 7 Mat, 7 A, 3 M, 1 Fec |
| *Serranus cabrilla* | Comber | Serranidae | LC | M | 74 | 7/8 | 35 LWR, 9 Sp, 8 G, 8 Diet, 7 A, 4 Mat, 3 M, 0 Fec |
| *Thunnus thynnus* | Atlantic bluefin tuna | Scombridae | EN | VH | 73 | 8/8 | 16 Diet, 14LWR, 14 G, 10 Sp, 8 A, 5 Mat, 3 M, 3 Fec |

*Note*: The commercial value (Val) is shown as price category (VH: very high; H: high; M: medium; L: low) and the protection status (IUCN) as IUCN Red List status category (LC: least concern; EN: endangered; DD: data deficient; NE: not evaluated; NT: near threatened; VU: vulnerable; CR: critically endangered). We included all the species with 8/8 and 7/8 characteristics studied. For the complete table, the reader may refer to the (S1 Table).

LWR: length-weight relationships; G: growth parameters; A: lifespan; Mat: length at maturity; Sp: onset and duration of spawning; Fec: fecundity; M: mortality; Diet: feeding preferences.

*luridus* and Marbled spinefoot *Siganus rivulatus*), as there is available information on at least seven biological characteristics for these eight species. With respect to the Mediterranean fish species dwelling in deep-sea habitats, 45% of them have zero available information on their biological characteristics, while only 24% have recorded data on at least half of the studied characteristics (Fig 3).

The available information on the previously least studied fish species in the Mediterranean Sea (Table 2 in [1]), has increased, as many of these species have had new records on their biological characteristics published within the last five years (Table 5).

The highest ratio of records per metric ton of production was observed for the velvet belly, *Etmopterus spinax* (5.1 records/t), longnose spurdog, *Squalus blainville* (4.6 records/t), bigscale sand smelt, *Atherina boyeri* (4.4 records/t) and Mediterranean sand smelt, *Atherina hepsetus* (2.2 records/t). The highest number of records to somatic length ratio was observed for the European pilchard, *Sardina pilchardus* (4.6 records/cm), red mullet *Mullus barbatus barbatus* (4.3 records/cm), common two-banded seabream, *Diplodus vulgaris* (3.1 records/cm), European anchovy, *Engraulis encrasicolus* (2.4 records/cm), picarel, *Spicara smaris* (2.2 records/cm) and common pandora, *Pagellus erythrinus* (2.1 records/cm).

Concerning the Mediterranean fish species with at least one studied biological characteristic and with available data on catches, maximum length and trophic level, the number of records had a moderately positive correlation to total catches (n = 125, Pearson r = 0.51, P < 0.001), and a very weak positive correlation to maximum reported length (n = 451, Pearson ρ = 0.17, P < 0.001). In contrast to total catches and maximum somatic length, the trophic level did not correlate with the number of records (n = 452, Spearman ρ = 0.09, P = 0.068) (Fig 4).

**Table 3. List of all the Mediterranean chondrichthyan species with studies on their biological characteristics.**

| Species | Common Name | Family | IUCN | Commercial Value | Number of Rec. | Number of Char. | Number of records per characteristic_ Present study |
|---|---|---|---|---|---|---|---|
| *Raja clavata* | Thornback ray | Rajidae | NT | M | 55 | 7 | 18 LWR, 12 Diet, 8 G, 6 A, 6 Mat, 3 Fec, 2 Sp |
| *Galeus melastomus* | Blackmouth catshark | Pentanchidae | LC | H | 54 | 6 | 18 Diet, 15 Mat, 10 LWR, 8 Sp, 2 Fec, 1 G |
| *Scyliorhinus canicula* | Lesser spotted dogfish | Scyliorhinidae | LC | M | 45 | 6 | 16 Diet, 12 LWR, 8 Mat, 6 Fec, 2 Sp, 1 G |
| *Squalus blainville* | Longnose spurdog | Squalidae | DD | M | 37 | 7 | 9 LWR, 9 Diet, 7 Mat, 5 G, 3 Sp, 2 A, 2 Fec |
| *Torpedo marmorata* | Marbled electric ray | Torpedinidae | DD | NA | 34 | 7 | 14 LWR, 6 Diet, 5 Mat, 4 Fec, 3 Sp, 1 G, 1 A |
| *Raja radula* | Rough ray | Rajidae | EN | M | 15 | 7 | 7 LWR, 3 Diet, 1 G, 1 A, 1 Sp, 1 Mat, 1 Fec |
| *Tetronarce nobiliana* | Electric ray | Torpedinidae | DD | NA | 15 | 6 | 8 LWR, 2 A, 2 Diet, 1 G, 1 Sp, 1 Fec |
| *Raja polystigma* | Speckled ray | Rajidae | LC | NA | 13 | 7 | 4 Diet, 3 LWR, 2 Fec, 1 G, 1 A, 1 Sp, 1 Mat |
| *Mustelus punctulatus* | Blackspotted smooth hound | Triakidae | DD | M | 12 | 5 | 5 Diet, 2 LWR, 2 Sp, 2 Mat, 1 Fec |
| *Dalatias licha* | Kitefin shark | Dalatiidae | VU | M | 11 | 4 | 6 Diet, 3 LWR, 1 Sp, 1 Mat |
| *Dasyatis marmorata* | Marbled stingray | Dasyatidae | DD | NA | 10 | 6 | 4 LWR, 2 Mat, 1 G, 1 A, 1 Fec, 1 Diet |
| *Leucoraja melitensis* | Maltese ray | Rajidae | CR | NA | 5 | 3 | 3 LWR, 1 Sp, 1 Fec |
| *Alopias vulpinus* | Thresher | Alopiidae | VU | H | 2 | 2 | 1 LWR, 1 Diet |
| *Cetorhinus maximus* | Basking shark | Cetorhinidae | EN | L | 1 | 1 | 1 LWR |
| *Glaucostegus halavi* | Halavi ray | Glaucostegidae | CR | NA | 1 | 1 | 1 LWR |
| *Carcharhinus longimanus* | Oceanic whitetip shark | Carcharhinidae | CR | M | 0 | 0 | |
| *Pristis pectinata* | Smalltooth sawfish | Pristidae | CR | M | 0 | 0 | |
| *Pristis pristis* | Common sawfish | Pristidae | CR | M | 0 | 0 | |
| *Dipturus batis* | Blue skate | Rajidae | CR | M | 0 | 0 | |
| *Sphyrna lewini* | Scalloped hammerhead | Sphyrnidae | CR | M | 0 | 0 | |
| *Sphyrna mokarran* | Great hammerhead | Sphyrnidae | CR | H | 0 | 0 | |

*Note*: The commercial value (Val) is shown as price category (VH: very high; H: high; M: medium; L: low) and the protection status (IUCN) as IUCN Red List status category (LC: least concern; EN: endangered; DD: data deficient; NE: not evaluated; NT: near threatened; VU: vulnerable; CR: critically endangered). For the complete table, the reader may refer to the (S2 Table).

LWR: length-weight relationships; G: growth parameters; A: lifespan; Mat: length at maturity; Sp: onset and duration of spawning; Fec: fecundity; M: mortality; Diet: feeding preferences).

## Discussion

Since the publication of the first gap analysis in 2017 [1], the research effort regarding the knowledge gaps in biological characteristics of Mediterranean marine fish species has intensified. The results showed that during the last five years (2016–2021) the knowledge gap on the biology of Mediterranean marine fishes was slightly narrowed as the percentage of fish species without any biological characteristics has dropped from 43% in 2016 [1] to 37% in 2021 (Fig 1 and Table 5). With respect to the number of records per species, the most studied species remain the same as in the previous gap analysis (Table 2). The general rule appears to be that species of high commercial value are also more likely to have been studied across their biological characteristics (Tables 2 and 5), reflected in the missing knowledge regarding the Gobiidae, Blennidae, Myctophidae, Gobiesocidae and Syngnathidae families (Table 4). The commercial status of the species belonging in the above families is described, mostly, as "NA" in FishBase [42].

Prompted by the original gap analysis, several journal articles were published aiming to reduce the identified knowledge gap. Two of these articles provided the first published diet

**Table 4. List of selected least studied families of non-commercial fish species in the Mediterranean Sea based on the number of studied characteristics and the number of records per characteristic.**

| Species | Common Name | Family | IUCN |
|---|---|---|---|
| *Lipophrys pholis* | Shanny | Blenniidae | LC |
| *Omobranchus punctatus* | Muzzled blenny | Blenniidae | LC |
| *Apletodon incognitus* | (clingfish) | Gobiesocidae | LC |
| *Diplecogaster bimaculata* | Two-spotted clingfish | Gobiesocidae | LC |
| *Benthophilus stellatus* | Stellate tadpole-goby | Gobiidae | LC |
| *Buenia lombartei* | (goby) | Gobiidae | NA |
| *Chromogobius quadrivittatus* | Chestnut goby | Gobiidae | LC |
| *Coryogalops ocheticus* | (goby) | Gobiidae | EN |
| *Eretmophorus kleinenbergi* | (morid cod) | Moridae | LC |
| *Guttigadus latifrons* | (morid cod) | Moridae | NA |
| *Diogenichthys atlanticus* | Longfin lanternfish | Myctophidae | LC |
| *Gonichthys cocco* | Cocco's lanternfish | Myctophidae | LC |
| *Entelurus aequoreus* | Snake pipefish | Syngnathidae | NA |
| *Hippocampus fuscus* | Sea pony | Syngnathidae | NA |

*Note*: The commercial value (Val) is shown as price category (VH: very high; H: high; M: medium; L: low) and the protection status (IUCN) as IUCN Red List status category (LC: least concern; EN: endangered; DD: data deficient; NE: not evaluated; NT: near threatened; VU: vulnerable; CR: critically endangered). For all the species listed in this table, commercial value is not known ("NA") and both the number of records and characteristics are equal to 0. For the complete table, the reader may refer to the (S3 Table).

records for the species phaeton dragonet *Synchiropus phaeton* [45] and yellowspotted puffer *Torquigener flavimaculosus* [46]. It is explicitly stated, in these publications, that the aim of the study was to reduce the knowledge gap, a notion that is also reflected, even if not explicitly stated, among other papers gathered for this study. Nevertheless, a considerable knowledge gap still exists as 37% of the Mediterranean marine fish species have no available information on their biological characteristics and 13% of them have very little information available regarding mostly the length-weight relationship trait.

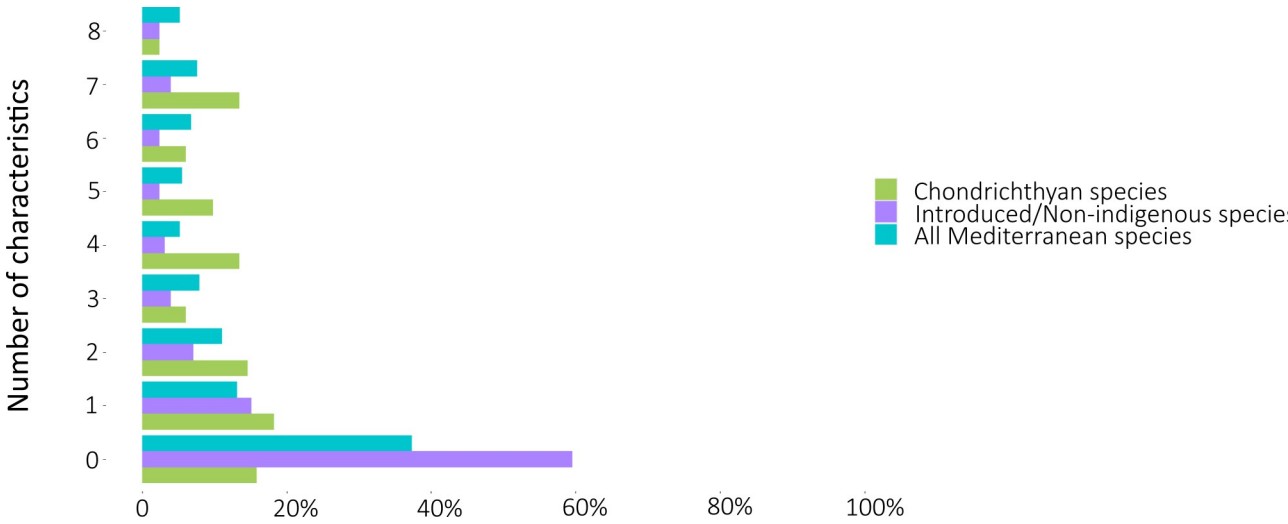

**Fig 2. Percentages of Mediterranean fish species with studied biological characteristics (0–8).** Blue colour: all Mediterranean species; purple colour: introduced/non-indigenous species; green colour: chondrichthyan species).

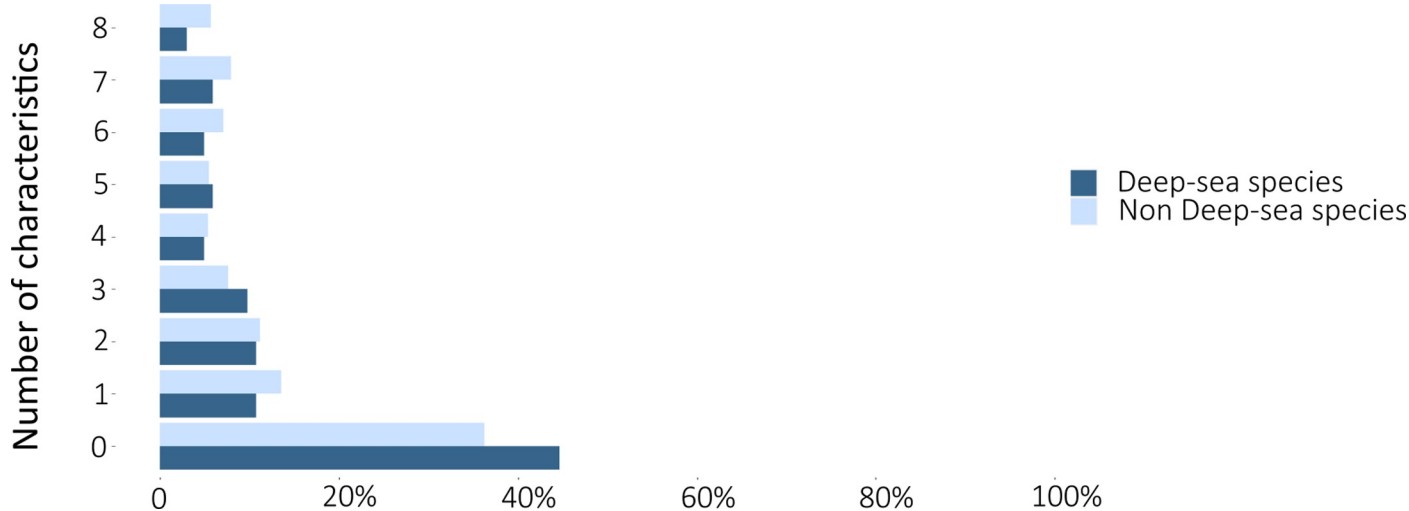

**Fig 3. Percentages of deep-sea and non deep-sea Mediterranean fish species with studied biological characteristics (0–8).** Dark blue colour: deep-sea species; light blue colour: non-deep-sea species).

Length-weight relationships (LWR) remain the most studied biological characteristic [1], as expected from the low cost and little effort and time required for its study [47, 48]. Contrary to the recommendations provided in Dimarchopoulou *et al.* (2017) [1] towards filling the gaps in less popular characteristics, papers published in the past five years, that focus on narrowing the knowledge gap, have been massively producing LWR records in a single publication [49, 50]. At the same time, information on other biological characteristics that can be cumbersome to obtain are harder to come across [51]. To this day, fecundity of marine Mediterranean fish species, remains one of the least studied characteristics.

On the other hand, maximum age, which was reported as "oddly missing" in the previous analysis, seems to have gained quite a few new records (n = 187; Table 1) in the last five years, for instance, for the species *Sciaena umbra* [52], *Trachinotus ovatus* [53] and *Alepes djedaba* [54]. Conversely, mortality records are still very few and exist for only a small percentage of the Mediterranean fish species (Fig 1, top row and Table 1). As stated in the initial gap analysis "*the maximum age and natural mortality of a population should be reported in all articles on growth*" [1], as maximum age is measured regardless, while natural mortality may be empirically estimated from growth parameters and sea temperature [55, 56].

In the first gap analysis it was also proposed that maximum economy of sampling should be applied, and the biological material should be thoroughly studied across their biological characteristics, specifically in cases of protected or vulnerable species [1]. Nevertheless, there is an issue emerging from this, as working with a single specimen produces sparse data with "*very little potential for analysis*" and a "*limited insight into its biology*" [51]. However, a recently developed approach [57] can be used to estimate LWR for data-poor species, based on a Bayesian hierarchical method. This method categorizes species in body shape groups and uses the prior knowledge and existing LWR studies of well-studied species to derive species-specific LWR parameters for data-deficient species that belong to the same group. The above-mentioned method was recently used to estimate the parameters of LWR of uncommon Mediterranean sharks and rays, utilizing even single records, thus making the examination of single records valuable [58].

The knowledge gap in the chondrichthyan life-history traits has been acknowledged [17] and efforts have been made for it to be narrowed (Tables 3 and 5). Reviews have been

**Table 5. Comparison between current and previously published records on the biology of selected Mediterranean fish species from Dimarchopoulou et al. (2017) [1], based on the number of studied characteristics and the number of records per characteristic.**

| Species | Common Name | Family | IUCN | Commercial Value | Number of Rec._ Present study | Number of Rec._ Dimarchopoulou etal. 2017 [1] | Number of Char._ Present study | Number of Char._ Dimarchopoulou etal. 2017 [1] | Number of records per characteristic_ Present study | Number of records per characteristic_ Dimarchopoulou *et al.* (2017) [1] |
|---|---|---|---|---|---|---|---|---|---|---|
| **Species with commercial value** | | | | | | | | | | |
| *Epinephelus costae* | Goldblotch grouper | Serranidae | DD | VH | **10** | 7 | 3/8 | 3/8 | **6 LWR**, 2 G, **2 Diet** | 4 LWR, 2 G, 1 Diet |
| *Argyrosomus regius* | Meagre | Sciaenidae | LC | M | **9** | 5 | **6/8** | 3/8 | 3 Sp, **2 LWR**, 1 G, **1 A**, 1 Mat, **1 Diet** | 3 Spawn, 1 G, 1 Mat |
| *Lepidotrigla dieuzeidei* | Spiny gurnard | Triglidae | LC | VH | **10** | 1 | **6/8** | 1/8 | **3 LWR**, **2 G**, **2 A**, **1 Sp**, **1 Mat**, **1 Fec** | 1 LWR |
| **Non-commercial species** | | | | | | | | | | |
| *Blennius ocellaris* | Butterfly blenny | Blenniidae | LC | NA | **10** | 8 | **3/8** | 2/8 | **7 LWR**, 2 Sp, **1 Diet** | 6 LWR, 2 Sp |
| *Callionymus lyra* | Dragonet | Callionymidae | LC | NA | 2 | 2 | 2/8 | 2/8 | 1 LWR, 1 Sp | 1 LWR, 1 Sp |
| **Species with atypical life strategies** | | | | | | | | | | |
| *Squatina aculeata* | Sawback angelshark | Squatinidae | CR | M | **16** | 3 | **5/8** | 3/8 | **6 LWR**, 3 Sp, 3 Mat, 3 Fec, **1 Diet** | 1 Sp, 1 Mat, 1 Fec |
| *Aetomylaeus bovinus* | Bull ray | Myliobatidae | CR | M | **11** | 1 | **6/8** | 1/8 | **4 LWR**, **2 Sp**, **2 Diet**, **1 G**, **1 A**, 1 Fec | 1 Fec |
| *Pterois miles* | Devil firefish | Scorpaenidae | LC | NA | **12** | 0 | **7/8** | 0/8 | **3 Diet**, **2 LWR**, **2 Sp**, **2 Mat**, **1 G**, **1 A**, **1 Fec** | |
| **Protected species** | | | | | | | | | | |
| *Mobula mobular* | Devil fish | Mobulidae | EN | NA | **4** | 0 | **1/8** | 0/8 | **4 LWR** | |
| *Carcharodon carcharias* | Great white shark | Lamnidae | VU | L | **5** | 0 | **2/8** | 0/8 | **3 Diet**, **2 LWR** | |

*Note*: The commercial value (Val) is shown as price category (VH: very high; H: high; M: medium; L: low) and the protection status (IUCN) as IUCN Red List status category (LC: least concern; EN: endangered; DD: data deficient; NE: not evaluated; NT: near threatened; VU: vulnerable; CR: critically endangered). The differences between the results of the present study and the Dimarchopoulou *et al.* (2017) [1] study are highlighted in bold. For the complete table, the reader may refer to the (S4 Table).

LWR: length-weight relationships; G: growth parameters; A: lifespan; Mat: length at maturity; Sp: onset and duration of spawning; Fec: fecundity; M: mortality; Diet: feeding preferences

published in the last five years regarding the large and overlooked group of chondrichthyans. For example, a significant gap in the length-weight relationships of 46 uncommon sharks and rays has been filled in using the valuable information from single records or few individuals and producing values reported for the first time at a global scale [58]. A number of new projects and studies showcase the rigorous effort to fill in the knowledge gap in the chondrichthyan fish species of the Mediterranean region. Boldrocchi *et al.* (2017) [59] have gathered bibliography on the feeding ecology of the elusive great white shark (*Carcharodon carcharias*). Geraci *et al.* (2021) [60] have published an extensive review on life history traits of the batoids (rays) in the Strait of Sicily. In the paper by Karachle *et al.* (2020) [51] there is a literature review of

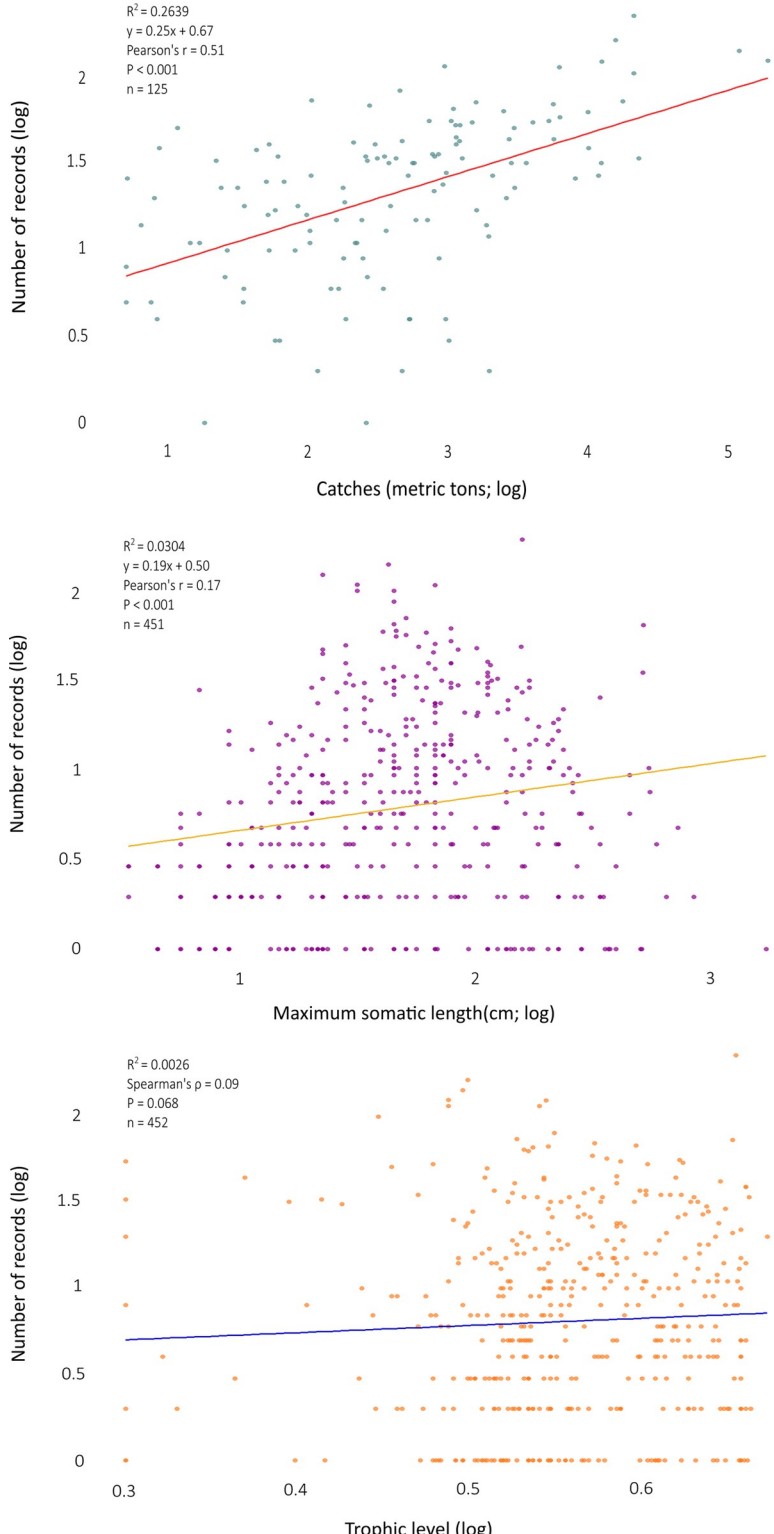

**Fig 4. The relationship between the number of records and total catches (in metric tons), maximum reported somatic length (cm) and trophic level, for the Mediterranean fish species with at least one studied biological characteristic and with available data on total catches (n = 125), length measurements (n = 451) and calculated trophic level (n = 452).** All variables were logarithmically transformed to improve visibility of the figure.

the near threatened sharpnose sevengill shark (*Heptranchias perlo*). In addition to these reviews, there is also a new data collection project, the Mediterranean Large Elasmobranchs Monitoring Program (MEDLEM), aiming to contribute to biodiversity management and conservation of the Mediterranean basin [61]. Chondrichthyans of the Mediterranean Sea have been described as "Endangered", "Vulnerable" and "Data deficient" by the IUCN Shark Specialist Group [17]. Data Deficient species are excluded from regional priority species lists, which impedes a much-needed status improvement for these species [18]. Although there have been attempts to predict the conservation status of Data Deficient chondrichthyan species through models [18], data collection for this group of species should still be a top priority for future research effort.

Another group of vulnerable Mediterranean fish species lacking data, is the deep-sea species group (Fig 3). Deep-sea fish show remarkable adaptations to this low productivity, low temperature and high-pressure environment [62]. These environmental conditions are tied to fish life history characteristics such as slow growth, delayed maturity and high maximum age [24], which affects negatively their resilience against fishing pressure [5, 63]. Our results confirm the findings of other studies [23] stressing out the knowledge gap regarding this particular group of fish in the Mediterranean region. As our view on these species remains obscure, exploitation of deep-sea fish stocks [64] and other deep-sea services intensifies [27], causing habitat disturbance, biodiversity loss [26, 65] and exponential decline in deep-sea functioning [66]. The Marine Strategy Framework Directive (MSFD 2008/56/EC) was EU's Integrated Maritime Policy plan to reach a good environmental status in the EU's marine ecosystems and to protect marine socio-economic resources. Although at first, the implementation of this directive was focusing on coastal areas, currently, it is migrating towards the deep Mediterranean Sea, to establish an ecosystem-based approach for its management. The approach is based on 11 qualitative descriptors, with the first one being "*biodiversity*" and requiring knowledge over "*the population demographic characteristics of the species*", meaning maximum age and fecundity, among other parameters [67]. Bridging the knowledge gap in Mediterranean deep-sea species will help in the establishment of a management strategy for the protection of the deep basin, and the eco-friendly blue growth of the Mediterranean countries [22, 68, 69].

Biological invasions from the Red Sea and the Indo-Pacific Ocean are a well-documented phenomenon [70], with the influx of species from the Suez Canal growing each year [34] and threatening Mediterranean biodiversity. This threat is amplified by climate change, as the thermophilic species arriving from the Red Sea spread faster due to the rising water temperature [71, 72]. The impacts that non-indigenous species have on local ecosystems vary from preying on native species and competing against them over the natural resources [39], to population losses and extinctions on a local scale [73]. Non-indigenous species impacts also affect societies on a socioeconomic level, as they introduce toxins and pathogens to the food webs, thus threatening aquaculture and fisheries and consequently human health [36, 39]. The current study showed that non-indigenous fish species are significantly less studied than any other group in this paper (chondrichthyans, deep-sea species or the whole of Mediterranean marine fish species; Figs 2 and 3). Although there have been several papers and reports that detail the sightings of non-indigenous fish species in the Mediterranean Basin [74], the effort to document their biological characteristics doesn't appear to be as intense, probably due to the laborious nature of collecting data on biological characteristics. Nevertheless, it would be a good first step if, at least LWR was documented for the non-indigenous fish species of the Mediterranean, as this is the least demanding biological characteristic to obtain [47, 48]. The incomplete, inaccurate and uncertain data on these species' biological characteristics, abundances, distributions etc., hinder any management attempt [34, 41].

It seems that an ongoing, simultaneous effort is being made regarding species of the Mediterranean marine biodiversity once neglected. The Mediterranean international trawl survey (MEDITS) programme [75], the EU Biodiversity Strategy for 2030 [76], the Mediterranean Biodiversity Records [70], frequent stock assessments [29, 77] and even citizen science projects [78, 79] are indicative of the newfound interest and intensive effort of scientists, stakeholders and the public towards discovering, monitoring and managing marine biodiversity in the Mediterranean region. Moreover, during record collection we also came across the data paper by Albouy *et al.* (2015) [80] which contains extensive information on biological characteristics, functional traits and the phylogeny of Mediterranean marine fishes, as well as their current and projected distributions. Although this is undoubtedly a valuable source of data, such papers and datasets containing raw data do not lie within the scope of this work and were therefore not included in our analysis. There are indeed multiple sources of information that could help fill in the gap of knowledge, but in the initial and current gap analysis, we focused on information that was derived from published papers found in FishBase [42], the largest database for fish, and the global scientific search engine SCOPUS.

To prioritize the actions needed to be taken, is key to any successful management strategy [81], especially when resources for management are limited [82]. In character with effectively targeting the identified gaps, the geographical and temporal aspect should also be taken into consideration. There have been efforts to identify existing data gaps of all Mediterranean marine databases, in an attempt to use them as a guide in the prioritization process [83]. The biological characteristics of fish species can be useful in this process, for example, data regarding the spawning period of species could provide temporal insight. Additionally, the subregional patterns detected would give spatial insight, as they revealed that the western and eastern Mediterranean are leading forces in biological data collection (Fig 1. *bottom row*), while the central part of the Mediterranean is sparse on information.

## Conclusions

The demand for an informed and unanimously shared management strategy of the Mediterranean Sea is urgent, as the understudied aspects of the its biodiversity need to be revisited within the framework of climate change, overfishing and the constant influx of new invasive species from the Red Sea [84]. Basic biological data are essential for future models, projections and scenario testing, priority species lists, and vulnerability assessments, all of which facilitate the understanding of the functioning of Mediterranean ecosystems and their response to change and can, therefore, inform ecosystem-based planning and management for the basin as a whole. Hence, it is encouraging that the gap in the biological knowledge of Mediterranean marine fish species has been reduced during the past few years, with new studies producing first records of species characteristics in the Mediterranean area and even globally.

In this pursue of data and knowledge, we recommend that the scientific society prioritizes identified gaps on key species over the well-studied commercial species, namely chondrichthyans and deep-sea fish species, as well as the infamous non-indigenous fish species. Additionally, the expansion of citizen science projects is commendable both for raising public awareness on ecological issues and for producing truly useful data [85, 86]. Finally, employing new methods of producing data through non-traditional sampling [3, 57, 58] is a practice aligned with the spirit of economy and efficacy, which is a notion embraced by both the initial and the present gap analysis.

## Supporting information

**S1 Table. List of the most studied fish species in the Mediterranean Sea based on the number of records (Number of Rec.), the number of studied characteristics (Number of Char.) and the number of records per characteristic (LWR: Length-weight relationships; G: Growth parameters; A: Lifespan; Mat: Length at maturity; Sp: Onset and duration of spawning; Fec: Fecundity; M: Mortality; Diet: Feeding preferences).** The commercial value (Val) is shown as price category (VH: very high; H: high; M: medium; L: low) and the protection status (IUCN) as IUCN Red List status category (LC: least concern; EN: endangered; DD: data deficient; NE: not evaluated; NT: near threatened; VU: vulnerable; CR: critically endangered). We included all the species with 8/8 and 7/8 studied characteristics and only the species with 6/8 studied characteristics and more than 40 records.
(DOCX)

**S2 Table. List of all the Mediterranean chondrichthyan species with or without studies on their biological characteristics.** (LWR: length-weight relationships; G: growth parameters; A: lifespan; Mat: length at maturity; Sp: onset and duration of spawning; Fec: fecundity; M: mortality; Diet: feeding preferences). The commercial value (Val) is shown as price category (VH: very high; H: high; M: medium; L: low) and the protection status (IUCN) as IUCN Red List status category (LC: least concern; EN: endangered; DD: data deficient; NE: not evaluated; NT: near threatened; VU: vulnerable; CR: critically endangered).
(DOCX)

**S3 Table. List of some of the least studied families of non-commercial fish species in the Mediterranean Sea based on the number of studied characteristics and the number of records per characteristic.** The commercial value (Val) is shown as price category (VH: very high; H: high; M: medium; L: low) and the protection status (IUCN) as IUCN Red List status category (LC: least concern; EN: endangered; DD: data deficient; NE: not evaluated; NT: near threatened; VU: vulnerable; CR: critically endangered). For all the species of this table, commercial value is not known ("NA") and both the number of records and characteristics are equal to 0.
(DOCX)

**S4 Table. Comparison between current and previously published records on the biology of selected Mediterranean fish species (Dimarchopoulou *et al.*, 2017) [1], based on the number of studied characteristics and the number of records per characteristic (LWR: Length-weight relationships; G: Growth parameters; A: Lifespan; Mat: Length at maturity; Sp: Onset and duration of spawning; Fec: Fecundity; M: Mortality; Diet: Feeding preferences).** The commercial value (Val) is shown as price category (VH: very high; H: high; M: medium; L: low) and the protection status (IUCN) as IUCN Red List status category (LC: least concern; EN: endangered; DD: data deficient; NE: not evaluated; NT: near threatened; VU: vulnerable; CR: critically endangered). The differences between the results of the present study and the Dimarchopoulou *et al.* (2017) [1] study are highlighted in bold.
(DOCX)

**S1 Appendix. The number of records per characteristic per species gathered from literature published from 2015 to 2021 and the total number of records per characteristic per species up until 2021.**
(XLSX)

**S2 Appendix. List of literature sources screened in the present study.**
(XLSX)

## Acknowledgments

We would like to thank the two reviewers for their comments which have greatly improved the manuscript.

## Author Contributions

**Conceptualization:** Eva Daskalaki, Donna Dimarchopoulou, Athanassios C. Tsikliras.

**Data curation:** Eva Daskalaki, Evangelos Koufalis.

**Formal analysis:** Eva Daskalaki, Evangelos Koufalis.

**Methodology:** Eva Daskalaki, Donna Dimarchopoulou.

**Resources:** Athanassios C. Tsikliras.

**Supervision:** Athanassios C. Tsikliras.

**Writing – original draft:** Eva Daskalaki.

**Writing – review & editing:** Evangelos Koufalis, Donna Dimarchopoulou, Athanassios C. Tsikliras.

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
