## [Decision Letter · Decision Letter 0]

12 Sep 2022

PONE-D-22-21777Scientific progress made towards bridging the knowledge gap in the biology of Mediterranean marine fishesPLOS ONE

Dear Dr. Daskalaki,

Thank you for submitting your manuscript to PLOS ONE. After careful consideration, we feel that it has merit but does not fully meet PLOS ONE’s publication criteria as it currently stands. Therefore, we invite you to submit a revised version of the manuscript that addresses the points raised during the review process.

In particular follow the requests of ref. n.2 about data valilability as an Appendix and about the details on methodology. 

We look forward to receiving your revised manuscript.

Kind regards,

Roberta Cimmaruta, PhD

Academic Editor

PLOS ONE

Journal Requirements:

"This research was partly funded by the European Union’s Horizon 2020 Research and Innovation Program (H2020-BG-10-2020-2), grant number No. 101000302 - EcoScope (Ecocentric management for sustainable fisheries and healthy marine ecosystems)."

"This research was partly funded by the European Union’s Horizon 2020 Research and Innovation Program (H2020-BG-10-2020-2), grant number No. 101000302 - EcoScope (Ecocentric management for sustainable fisheries and healthy marine ecosystems)."

"This research was partly funded by the European Union’s Horizon 2020 Research and Innovation Program (H2020-BG-10-2020-2), grant number No. 101000302 - EcoScope (Ecocentric management for sustainable fisheries and healthy marine ecosystems)."

Reviewers' comments:

Reviewer's Responses to Questions

**Comments to the Author**

1. Is the manuscript technically sound, and do the data support the conclusions?

Reviewer #1: Yes

Reviewer #2: Yes

2. Has the statistical analysis been performed appropriately and rigorously? 

Reviewer #1: Yes

Reviewer #2: N/A

3. Have the authors made all data underlying the findings in their manuscript fully available?

Reviewer #1: Yes

Reviewer #2: No

4. Is the manuscript presented in an intelligible fashion and written in standard English?

Reviewer #1: Yes

Reviewer #2: Yes

5. Review Comments to the Author

Reviewer #1: In the paper Scientific progress made towards bridging the knowledge gap in the biology of Mediterranean marine fishes, Daskalaki and colleagues provided an update of a previous analysis conducted in 2017, aimed to uncover gaps in the current knowledge on the biology of Mediterranean marine fishes.

The authors used mainly data from FishBase and scientific literature, showing how the current knowledge in the last 5 years only shrunk by 6%. However, some interesting results emerge from a situation still far from being optimal. For instance, they found how 40 new species have at least one study on their biology and that scientific research has slightly shifted towards traditionally neglected species, as for instance sharks, rays and chimaeras. Overall, their findings suggest an increasing awareness in the study of fish biology in the Mediterranean Sea not exclusively devoted to commercial fishes.

I did not find major concerns in this paper, as it is well written, concise and goes straight to the focus of their main questions.

However, I am wonder if the Authors are aware of the paper of Albouy and collegues, FishMed: traits, phylogeny, current and projected species distribution of Mediterranean fishes, and environmental data, Ecology 96(8) 2312, were the authors provided an updated overview (and datasets) of traits, phylogeny and current and projected spatial distribution of 635 Mediterranean fish species, “compiled from a published expert knowledge atlas of fishes of the northern Atlantic and the Mediterranean (FNAM) edited between 1984 and 1986 and from an updated exotic fish species list”. I understand that the final aim of this paper is to update the actual knowledge following the same procedure used in a recently published paper from the authors. At the same time, I am aware how FishBase still represents one (if not the most) complete and up-to-date global biodiversity information system on finfishes. Anyway, it would be interesting to understand if and how the data provided in this dataset (including the literature sources) are able or not to add information on the gap analysis of Mediterranean fish biology.

When working with non-proprietary data, as for instance in meta-analysis, one should be aware of the multiple and often different sources of information that, sometimes, help filling the gap of knowledge. For instance, Courtney et al. (2011) Errors in Length-weight Parameters at FishBase.org. Nature Precedings (2011). https://doi.org/10.1038/npre.2011.5927.1, cast doubts on how specific parameters (as for instance one used in the current ms, that is, length-weight) are

not generally reliable. Of course, the above-mentioned paper mainly used non-marine fish species, but the results obtained still cast serious doubts on the reliability of some parameters for other species. Such reflection applies, obviously, for those parameters shared in both databases.

Besides my comments above, which I hope can stimulate a deep discussion about the use of different (and often fragmented) information available on different databases that might influence the knowledge gap in the biology of Mediterranean marine fishes

I did not find major issues on the ms.

Reviewer #2: This study is an updated version of a previous study (Dimarchopoulou et al. 2017) aiming to quantify knowledge gaps about the traits of Mediterranean fishes. This study is quite similar to the original one but given the time passes since the original publication is still useful information that needs to be put out there. The manuscript is generally well written and I have few comments. The two major ones are:

1. To make this a truly useful paper, it is important to actually add the data on the traits collected as a (appendix) table. This will be a tremendously useful resource for marine scientists in the region and I feel is well worth the effort.

2. Methodologically, more information is needed on how the trait were searched in Scopus (for those that were not found in FishBase). What were the search words exactly? How many results were retrieved and screened? How much data was added on top of that presented in FishBase?

If these comments are addressed I feel the manuscript should be accepted to this journal.

Other minor comments:

In “The lack of data regarding basic biological characteristics of many deep-sea species”……

I would replace “almost impossible” with a more moderate word.

Explain what is meant exactly for absolute and relative number of oocytes.

Explain what the diet categories were and how gut content analyses was matched to these categories.

“However, a recently developed method [56], with which LWR can be estimated even from single fish records, based on a Bayesian hierarchical method,” Bayesian mthods are related to statistics. Please explain in a few words what this method is based on in terms of data, not stats. No matter how fancy the stats, you cannot infer LWR from a single point.

The grammar is off in the first sentence of the conclusions.

Conclusions: “dreaded” conveys too much emotions. I would use a less emotional phrase.

I would add another figure showing the cumulative number of traits with records (as Fig. 5 in Dimarchopoulou et al. 2017) to help assess if the rate has changed since the last assessment.

6. PLOS authors have the option to publish the peer review history of their article (what does this mean?). If published, this will include your full peer review and any attached files.

Reviewer #1: No

Reviewer #2: **Yes: **Jonathan Belmaker

---

## [Author Response · Author response to Decision Letter 0]

14 Oct 2022

Response letter of Daskalaki et al. (PONE-D-22-21777) 

The authors would like to thank the two reviewers for their comments and suggestions that really improved this manuscript.

Reviewer #1

In the paper Scientific progress made towards bridging the knowledge gap in the biology of Mediterranean marine fishes, Daskalaki and colleagues provided an update of a previous analysis conducted in 2017, aimed to uncover gaps in the current knowledge on the biology of Mediterranean marine fishes. The authors used mainly data from FishBase and scientific literature, showing how the current knowledge in the last 5 years only shrunk by 6%. However, some interesting results emerge from a situation still far from being optimal. For instance, they found how 40 new species have at least one study on their biology and that scientific research has slightly shifted towards traditionally neglected species, as for instance sharks, rays and chimaeras. Overall, their findings suggest an increasing awareness in the study of fish biology in the Mediterranean Sea not exclusively devoted to commercial fishes.

I did not find major concerns in this paper, as it is well written, concise and goes straight to the focus of their main questions.

Comment 1

However, I am wonder if the Authors are aware of the paper of Albouy and collegues, FishMed: traits, phylogeny, current and projected species distribution of Mediterranean fishes, and environmental data, Ecology 96(8) 2312, were the authors provided an updated overview (and datasets) of traits, phylogeny and current and projected spatial distribution of 635 Mediterranean fish species, “compiled from a published expert knowledge atlas of fishes of the northern Atlantic and the Mediterranean (FNAM) edited between 1984 and 1986 and from an updated exotic fish species list”. I understand that the final aim of this paper is to update the actual knowledge following the same procedure used in a recently published paper from the authors. At the same time, I am aware how FishBase still represents one (if not the most) complete and up-to-date global biodiversity information system on finfishes. Anyway, it would be interesting to understand if and how the data provided in this dataset (including the literature sources) are able or not to add information on the gap analysis of Mediterranean fish biology.

Reply

Indeed, this is a very good point. We are familiar with the FishMed database, which we believe is a valuable source of information on Mediterranean fishes and an important tool for researchers. However, as the reviewer acknowledges, the scope of the present work was to include only published scientific output and update a previous review using the same methodology. Thus, we have not used raw biological data available in stock assessments (such as GFCM and STECF) and databases (e.g., FishMed, Med&BS RDBFIS, Ormef). We have now included a reference to FishMed to improve data availability to the readers and to provide a better description of the most complete datasets that may be used to examine hypotheses by scientists and complement their publications. 

Comment 2

When working with non-proprietary data, as for instance in meta-analysis, one should be aware of the multiple and often different sources of information that, sometimes, help filling the gap of knowledge. For instance, Courtney et al. (2011) Errors in Length-weight Parameters at FishBase.org. Nature Precedings (2011). https://doi.org/10.1038/npre.2011.5927.1, cast doubts on how specific parameters (as for instance one used in the current ms, that is, length-weight) are not generally reliable. Of course, the above-mentioned paper mainly used non-marine fish species, but the results obtained still cast serious doubts on the reliability of some parameters for other species. Such reflection applies, obviously, for those parameters shared in both databases.

Reply

Agreed and thanks for pointing this out. Indeed, some of the parameters that are routinely determined in scientific papers are based on non-realistic assumptions and result in errors. FishBase performs a quality check on most of the parameters listed in the database based on sample size, range of lengths/weights included in the sample, statistical tests and/or common practices in fisheries biology. When a parameter or a method violates any of the standards or deviates a lot from what it is expected then it is marked as questionable in the corresponding “Tables” of FishBase. For example, when growth parameters are to be included, they are considered questionable if t0 is lower than -1.5 y or if Linf deviates a lot from Lmax. We had already excluded questionable records from the first version of our analysis and we have now stated so in material and methods by adding the text: “…leaving out all growth records of “questionable” status in FishBase [42].”.

Reviewer #2

This study is an updated version of a previous study (Dimarchopoulou et al. 2017) aiming to quantify knowledge gaps about the traits of Mediterranean fishes. This study is quite similar to the original one but given the time passes since the original publication is still useful information that needs to be put out there. The manuscript is generally well written and I have few comments. The two major ones are: 

Major Comment 1

To make this a truly useful paper, it is important to actually add the data on the traits collected as a (appendix) table. This will be a tremendously useful resource for marine scientists in the region and I feel is well worth the effort.

Reply

Agreed. A new Appendix was added (Appendix S1), containing the number of records per characteristic per species gathered from literature published between 2015-2021 as well as the total number of records per characteristic per species up to 2021. 

Major Comment 2

Methodologically, more information is needed on how the trait were searched in Scopus (for those that were not found in FishBase). What were the search words exactly? How many results were retrieved and screened? How much data was added on top of that presented in FishBase?

Reply

Agreed. More information was added in Methods that now describes in detail the search process in SCOPUS. In addition, we added another appendix (Appendix S2) containing the papers extracted from FishBase and SCOPUS, that have been screened for this study.

Minor comment 1

In “The lack of data regarding basic biological characteristics of many deep-sea species”……

I would replace “almost impossible” with a more moderate word.

Reply

Agreed and rephrased.

Minor comment 2

Explain what is meant exactly for absolute and relative number of oocytes.

Reply

Agreed. The definitions on absolute and relative fecundity were added with references.

Minor comment 3

Explain what the diet categories were and how gut content analyses was matched to these categories.

Reply

Within the scope of this gap analysis we recorded the number of papers per biological characteristic per species -as it is shown in the newly added Appendix S1-. So, we did not divide diet preferences and gut content into categories. In order to clarify this we have added the following text in the manuscript: “Specifically, we recorded the existence (or not) of papers containing information on eight biological characteristics for every species in the list.”.

Minor comment 4

“However, a recently developed method [56], with which LWR can be estimated even from single fish records, based on a Bayesian hierarchical method,” Bayesian methods are related to statistics. Please explain in a few words what this method is based on in terms of data, not stats. No matter how fancy the stats, you cannot infer LWR from a single point.

Reply

Agreed and elaborated as proposed.

Minor comment 5

The grammar is off in the first sentence of the conclusions.

Reply

Agreed and corrected.

Minor comment 6

Conclusions: “dreaded” conveys too much emotions. I would use a less emotional phrase.

Reply

Agreed and rephrased.

Minor comment 7

I would add another figure showing the cumulative number of traits with records (as Fig. 5 in Dimarchopoulou et al. 2017) to help assess if the rate has changed since the last assessment.

Reply

Although it is a good addition to the study and valuable as information, unfortunately it cannot be done because Figure 5 in Dimarchopoulou et al. (2017) was a count of articles on specific characteristics of species, as they appeared in SCOPUS (i.e., articles on spawning of Mediterranean fishes per year) and does not include any information on which (new) species records had been published each year.

Additional note

Two references were added in the reference list [43, 80] in line with the reviewers’ comments. The Qadri et al. (2015) paper was added as reference to the definitions on absolute and relative fecundity. The Albouy et al. (2015) paper was added after the first comment of reviewer #1.

---

## [Editor Report · Decision Letter 1]

26 Oct 2022

Scientific progress made towards bridging the knowledge gap in the biology of Mediterranean marine fishes

PONE-D-22-21777R1

Dear Dr. Daskalaki,

We’re pleased to inform you that your manuscript has been judged scientifically suitable for publication and will be formally accepted for publication once it meets all outstanding technical requirements.

Kind regards,

Roberta Cimmaruta, PhD

Academic Editor

PLOS ONE